# Ordering sequential competitions to reduce order relevance: Soccer penalty shootouts

**Nils Rudi[1] *, Marcelo Olivares[2,3], Aditya Shetty[4]**

**1** Yale School of Management, New Haven, CT, United States of America, **2** Universidad de Chile, Santiago, Chile, **3** Instituto Sistemas Complejos de Ingenieria, Santiago, Chile, **4** Simon Business School, University of Rochester, Rochester, NY, United States of America

* nils.rudi@yale.edu

**Data Availability Statement:** The repository that stores all the data for replication is now active. The permanent link is: https://doi.org/10.34691/FK2/QEZXKG.

## Abstract

In sequential competitions, the order in which teams take turns may have an impact on performance and the outcome. Previous studies with penalty shootouts have shown mixed evidence of a possible advantage for the first shooting team. This has led to some debate on whether a change in the rules of the game is needed. This work contributes to the debate by collecting an extensive dataset of shootouts which corroborates an advantage for the first shooter, albeit with a smaller effect than what has been documented in previous research. To evaluate the impact of alternative ordering of shots, we model shootouts as a probability network, calibrate it using the data from the traditional ordering, and use the model to conduct counterfactual analysis. Our results show that alternating the team that shoots first in each round would reduce the impact of ordering. These results were in part developed as supplement to field studies to support the International Football Association Board's (IFAB) consideration of changing the shooting order.

## Introduction

In sequential competitions, the order in which teams take turns may have an impact on performance and outcomes. In the case of soccer penalty shootouts, it has been argued that the starting team has an advantage and hence that alternative orders should be considered. The natural approach to evaluate alternative orders would be to design and implement a field experiment. However, as we demonstrate, performing a field experiment of an alternative order with reasonable statistical power, even with full implementation including all the main competitions, would take four years. In settings such as ours, doing counterfactual analysis using data from the current policy is the <u>only</u> way to evaluate well informed decisions since field experiments are not practical. To support this policy decision of the International Football Association Board (IFAB), we developed a data driven model, calibrated with a large-scale dataset. The model is rigorous in terms of representing the key issues, but also sufficiently intuitive so that it can be understood and interpreted by key stakeholders. The resulting decision was not to go further with alternative orders at this point. Although this may seem a negative result, it is important to consider that in-action decisions are often even more important than action-

**Funding:** MO received funding from Agencia Nacional de Investigacion y Desarrollo (ANID), Chile, grants Fondecyt 1181201 and PIA/APOYO AFB180003. https://www.anid.cl/ The funders had no role in study design, data collection and analysis, decision to publish, or preparation of the manuscript.

**Competing interests:** The authors have declared that no competing interests exist.

decisions in that they prevent costly failures and free up resources for other initiatives and potential policy changes.

Soccer is the world's largest and most global sport. The modern rules of the game date back to the mid-19th century and have since only been subject to a few rare changes. Being a low-scoring sport, a little more than a quarter of matches are tied after the regular 90 nominal minutes. In league tournaments with a round-robin structure, this will in general be the final result of the match. But when a winner needs to be determined, mainly in elimination tournaments, the match will typically go to 30 nominal minutes extra time. Of these matches, close to half of them will still not have a winner determined and will in general go to a penalty shootout, where players from the two teams take alternating shots from the penalty mark. The shootout procedure is determined by Law #10 of the game (http://theifab.com/laws), and can be summarized by four parts: (1) It is determined which goal to use; (2) it is determined which team will go first—defined as Team A throughout the paper (with Team B going second); (3) the team with the most goals in the first 5 rounds wins the match; (4) if a winner is not determined after 5 rounds then the shootout will continue with sudden-death rounds—the winner will be the first team that scores in a round in which the other team does not. Until 2003, a fair coin was used to determine which would be Team A. Since 2003, the fair coin determines which of the two teams' captain gets to choose whether to be Team A or B.

While penalty shootouts have given some of the most iconic moments of soccer, the concept has also been the subject of controversy. One criticism has been that the fair coin—a factor external to the game—influences its outcome. Clearly, if the scoring rate of each shooter were independent of the order, any reordering within part (3) and part (4) would not change the probability of Team A winning the shootout—and the fair coin would not influence the outcome. However, [1], using data from 12 tournaments (hereon referred to as APH tournaments), report that Team A wins in 60.5% of the cases based on 129 pre-2003 shootouts and in 59.2% of the cases based on 269 shootouts that include post-2003 cases. In addition, they conducted surveys with players and coaches (both professional and amateur) who report a perceived advantage of shooting first: more than 90% would prefer to shoot first when given the opportunity to choose. [2] analyze the first-shooter advantage using a superset of pre-2003 matches with 540 shootouts from the APH tournaments, finding that Team A wins in 53.3% of them. Later, [3] further expands to 1001 shootouts up until 2012 and reports 60.6% of shootouts won by Team A. A more recent study by [4] uses a sample of 252 shootouts from French competitions and finds no significant advantage of Team A. Hence, there is disagreement regarding to what extent there is an advantage for Team A—both in terms of magnitude and statistical significance (the first and third of these find such support, while the second and fourth do not).

On June 15, 2017, IFAB and FIFA launched the play fair! initiative, probably the most radical initiative toward the rules of the game in its history. A major element of this initiative, classified as "ready for testing/experiments," was to consider alternating the shooting order between the rounds of penalty shootouts. In the current scheme, called ABAB, Team A shoots first in every round. A frequently proposed alternative is to alternate the rounds' first shooter, called ABBA since in the first round the order is AB, the second round BA, and so on. The two key elements of consideration for the ultimate decision of such a change were (i) to what extent it would lead to additional complexity for the operations of the shootout and for the spectators and (ii) to what extent it would reduce the influence of the fair coin. To address element (i), FIFA tested the alternative order in U17 and U20 tournaments and in the League Cup and Community Shield, with a total of 12 shootouts. To further support this evaluation, we performed a major data collection and rigorous analysis to address (ii), evaluating whether ABBA reduces the influence of the fair coin, the main focus of this paper.

For this purpose, we follow the approach of [5] and model the evolution of the penalty shootout as a network with stochastic transitions across states representing the outcome on each round of the shootout. [5] use this analytical model to show the impact of different types of psychological pressure; in contrast, we use it for evaluating counterfactuals of alternative orders, similar to the approach taken by [6]. A realistic evaluation of these counterfactuals requires specifying the transition probabilities between states. [6] calibrate their model using score probabilities that depend on the kick number, based on the empirical results of [1]. Our extensive data collection enabled us to calibrate a more detailed network model of the shootout, where the scoring probabilities are also affected by the score difference across teams, which shows up as an important factor affecting shot outcomes.

Subsequently, in their 133rd annual business meeting on November 22, 2018, IFAB, the governing body of the rules of soccer, decided not to further consider alternative orders for penalty shootouts. Beyond this specific setting, we contribute to the theory and practice of evaluating alternative ordering policies in sequential competitions when performance depends on it.

## Data and analysis

In its simplest form, estimating the impact of changing the penalty shooting order requires computing two numbers: the influence of the fair coin with the traditional ABAB order and the influence of the fair coin after a change to ABBA.

Given the controversy of the above-mentioned papers in estimating the influence of the fair coin with the current order, we took special care in collecting a representative sample of shootouts following an objective sampling procedure.

Step 1.   We collected all the games from worldfootball.net, one of the most comprehensive public data sources on football matches. The data were downloaded on December 2018 and dates back to 1970. From this sample, we identified all the games reported with the extension 'pso', which indicates that the game ended in a penalty shootout.

Step 2.   To make sure that the list of shootouts is complete, we checked external sources to find additional shootouts not included in worldfootball.net. There are many external sources and it was not feasible to check them for all competitions, so we focused in collecting an exhaustive sample for all the APH competitions, to facilitate comparison with previous work.

1. For each of the APH competitions we found the earliest season reported in worldfootball.net after which shootouts were introduced in the the competition.

2. For each season in the APH competitions, we counted the number of games and compared it with the games reported on other sources (Wikipedia, Linguasport, UEFA.com and Rec.Sport.Soccer Statistics Foundation). For the additional matches found in these sources, we added those that ended in penalty shootouts to the sample. This includes 33 shootouts for English Cups and 432 for Copa del Rey. Copa del Rey matches where incomplete in worldfootball.net before 2000, so most of this sample was collected from Linguasport, which covers from 1970 up to the 2016/2017 season. From season 2017/2018 onwards, the data for this competition were collected from worldfootball.net. We checked some of the shootouts reported in both sources and the information is consistent.

Step 3.   Using the same data sources, we collected information about the sequence of shots on each of the identified shootouts looking at the detailed match information. The

number of shootouts with missing sequence information was relatively small for all major competitions except for Copa del Rey (the main data source of that competition did not include the detailed information). This competition has the highest number of shootouts across the APH competitions and is one of the major drivers of the reported results in [3]. To complete these missing data, three research assistants (Spanish native speakers) were hired to manually collect data on shootout sequence from Spanish newspaper archives (Mundo Deportivo, ABC and others). From these articles, 172 additional shootouts were collected for Copa del Rey.

Table 1 summarizes the results of this data collection process. We provide detailed information for the 12 APH tournaments to facilitate comparison with previous work, but we also analyzed an extended sample with 65 additional competitions (reported in the last row of Table 1). The proportion of shootouts with sequence information varies considerably across competitions, therefore the results may not be representative of the population of APH competitions. To further validate this, the last column of the table shows the proportion of shootouts won by Team A. We used these proportions to run a chi-square test under the Null hypothesis that the probability of Team A winning is the same across competitions. The resulting p-value of this test is 0.72, suggesting there are no systematic differences in the first-mover advantage across competitions, although the statistical power of this test is low due to the relatively small sample size for each competition.

## Evidence of first shooter advantage

Table 2 reports the fraction of shootouts won by Team A. The summary statistics are grouped by tournaments between club teams (including international club tournaments) and between national teams. Results are calculated for the APH and the extended set of tournaments separately. The p-value of a binomial one-sided t-test is calculated to test if the fraction is significantly larger than 50%. In all the cases reported, about 55% of the shootouts were won by Team A, which is a 10% point advantage relative to Team B. Putting this in perspective, Team A will have a 22% higher probability of winning the shootout than Team B. Given the low

Table 1. Details of the data collection process for APH competitions and the extended set of competitions.

| | Step 1 | Step 2 | Step 3 | | |
|---|---|---|---|---|---|
| Competition | wf.net | other | sequence | % with seq | Team A won |
| African Cup | 30 | | 22 | 73% | 0.50 |
| Asian Cup | 21 | | 12 | 57% | 0.50 |
| Champions | 58 | | 51 | 88% | 0.63 |
| Copa America | 23 | | 20 | 87% | 0.65 |
| Copa del Rey | 12 | 432 | 185 | 42% | 0.58 |
| Cup Winners | 32 | | 29 | 91% | 0.59 |
| English Cups | 278 | 33 | 157 | 50% | 0.52 |
| European Champ. | 18 | | 18 | 100% | 0.39 |
| German Cups | 221 | | 205 | 93% | 0.50 |
| Gold Cup | 12 | | 11 | 92% | 0.55 |
| UEFA | 146 | | 132 | 90% | 0.52 |
| World Cup | 26 | | 26 | 100% | 0.58 |
| **Total APH data** | 877 | 465 | 868 | 65% | 0.54 |
| Other 65 competitions | 3119 | | 755 | 24% | 0.56 |
| **Total extended data** | 3996 | 465 | 1623 | 36% | 0.55 |

**Table 2. Summary statistics of shootouts for APH tournaments and the extended tournaments, showing the percent of matches where the starting team wins the shootout.**

| Type | APH tournaments | | | Extended tournaments | | |
|---|---|---|---|---|---|---|
| | n | Mean | p-value | n | Mean | p-value |
| Club teams | 759 | 53.75% | 0.0192 | 1474 | 54.82% | 0.0001 |
| National teams | 109 | 53.21% | 0.2525 | 149 | 53.69% | 0.1846 |
| Total | 868 | 53.69% | 0.0149 | 1623 | 54.71% | <0.0001 |

scoring rate of soccer, luck plays a strong role and to gain a similar advantage in regular play would require a substantial skill difference. For club teams, where the sample size is large, as well as when combining club teams and national teams, the p-values suggest that Team A has a winning probability above 50% at reasonable levels of statistical significance, in support of an advantage for the first shooting team.

In magnitude, our results are much closer to [2] than to [1] and [3], but our larger sample size enables a more precise estimation and higher statistical power to detect an advantage for Team A.

## A network model of penalty shootouts

Although the previous analysis at the shootout level suggests a significant advantage for Team A, it does not provide enough information about how changes to the current order would affect this advantage; hence a more detailed unit of analysis is needed. A desired unit of analysis would provide enough detail to evaluate the impact of ordering, but at the same time keep the analysis parsimonious. In this spirit, we formulate the evolution of shootouts as a probability network of states that capture round-wise score transitions between the teams. A state is defined by the combination of the score difference $s$ in favor of Team A at the beginning of round $t$. Transitions between states are defined by the change in the score difference from $s$ to $s + c$ between rounds $t$ and $t + 1$, captured by an ordered trinomial r.v. with probabilities $q_c(s, t)$, $c \in \{-1, 0, 1\}$. Fig 1 illustrates the network, using the extended dataset to calculate the possible round-wise score transitions (represented by gray arrows). The northbound arrows not pointing to any node are where Team A is determined the winner of the shootout, and the corresponding southbound arrows represent Team B being determined the winner.

A key assumption of this network model is that the probabilities determining the change in score difference from one round to the next depend only on the score differences between the two teams, and are independent of the path that lead to a given state. This Markovian property facilitate the evaluation of alternative orders, as we show in the next section. However, this assumption is not innocuous as it may rule out some psychological mechanisms that have been suggested in sports. For example, it rules out the Gambler's fallacy [7] which has been reported by [8, 9] and [10] in soccer, where goal-keepers are more likely to dive in the opposite direction after receiving penalty shots repeatedly in one direction. The Markov assumption also rules out a potential "hot-hand" phenomena (e.g. [11]), in which a shooting team or the keeper may exhibit streaks of success/failures across rounds. For this reason, we conducted some additional data analysis to test for the markov assumption, which are described at the end of this section.

Fig 1 shows additional statistics calculated with the extended dataset, which are useful to understand in further detail how the advantage of Team A evolves round-by-round. The pie charts on each node represent the proportion of shootout sample paths visiting the state. The thick solid arrows at each trinomial node (a node that has 3 possible transitions out of

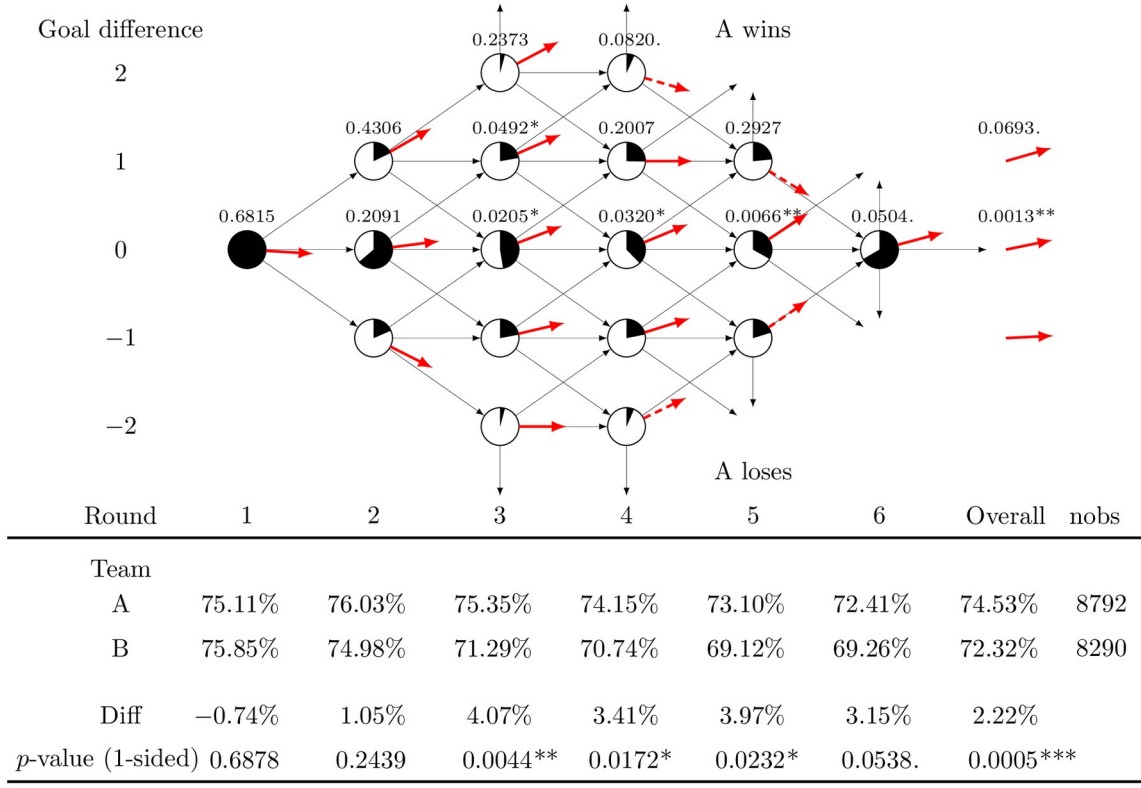

| Round | 1 | 2 | 3 | 4 | 5 | 6 | Overall | nobs |
|---|---|---|---|---|---|---|---|---|
| **Team** | | | | | | | | |
| A | 75.11% | 76.03% | 75.35% | 74.15% | 73.10% | 72.41% | 74.53% | 8792 |
| B | 75.85% | 74.98% | 71.29% | 70.74% | 69.12% | 69.26% | 72.32% | 8290 |
| Diff | −0.74% | 1.05% | 4.07% | 3.41% | 3.97% | 3.15% | 2.22% | |
| *p*-value (1-sided) | 0.6878 | 0.2439 | 0.0044** | 0.0172* | 0.0232* | 0.0538. | 0.0005*** | |

**Fig 1. Round-wise score transitions.** Summary statistics (using the extended dataset) of shot performance for first and second shooting team, by round. The p-values of alternative statistical tests are reported, with significance levels indicated by (.) 10%, (*) 5%, (**) 1% and (***) 0.1%.

it) provide a graphical representation of the drift of the score difference out of it, $\bar{q}(s,t) = q_1(s,t) - q_{-1}(s,t)$. Thick solid arrows pointing upward represent an advantage toward Team A, and we tested whether the advantages are statistically significant. Since some of the statistical tests are non-trivial, they are all based on bootstrapped samples constructed as follows: First, sample with replacement the same number of shootouts as in the dataset (880 for APH and 1635 for the extended tournaments). Second, using this re-sample, compute the transition probabilities in the network. Store these samples and estimates and repeat the steps for re-samples $r = 1, \ldots, 100000$. For each of the center nodes, $s = 0$, a one-sided t-test was used to test for an upward (positive) drift, in favor of Team A. The p-values of these tests are reported on top of each node (asterisks indicate the significance level at which the Null of zero drift can be rejected). All center nodes of rounds $t \geq 3$ show a significant positive drift in favor of the first shooting team. A similar test was run by grouping all the center nodes in a single test, with an average drift of 0.0251 and p-value 0.0013.

For the trinomial nodes in the upper portion of the network, with score difference $s \geq 1$ in favor of Team A, we compared the drift of each node $\bar{q}(s,t)$ with the drift in favor of Team B in the corresponding lower node of the network $-\bar{q}(-s,t)$. A one-sided t-test was run to evaluate the Null of equal drift against the alternative of a difference in drifts in favor of Team A. In general, the point estimates of the difference in drift suggest a larger drift for Team A, but only for $t = 3$, $|s| = 1$ is statistically significant (p-value = 0.0492, indicated over the node). A joint test for this difference in drift was run on rounds $t = 2, 3, 4$ for $|s| = 1$, with an average

drift difference of 0.0433 in favor of Team A (p-value = 0.0693), providing weak statistical evidence of an advantage for Team A when being ahead with a goal.

The outbound dashed arrows on some of the nodes (e.g., $s = 1$, $t = 5$) are from states limited to binomial transitions, where the only possible outcome in which the shootout continues is when the team ahead misses its shot and the lagging team scores. They represent the probability that the leading team <u>does not</u> win directly out of the state $q_{-sign(s)\cdot 1}(s, t)$. Thereby the shootout progresses to the next round, and hence it is not relevant to compare their slopes to those of the solid arrows. Similar tests were used to compare these upper nodes with the respective nodes in the bottom. Out of rounds 4 and 5, the tests only suggest a weak statistically significant advantage of Team A in round 4, with Team A having a 6.23% point higher probability than Team B of winning the shootout directly out of this state (p-value = 0.0820).

The analysis so far suggests a statistically significant advantage of Team A from rounds 3 onward when the score is tied. To further understand this first shooter advantage, the scoring proportions were calculated for teams A and B at each round of the shootout, reported in the lower part of Fig 1. In all rounds except the first one, Team A has a higher scoring proportion: the difference is small and not statistically significant in rounds 1 and 2, as suggested by the p-values of a one-sided test of proportions reported at the bottom of the figure, but becomes larger and statistically significant in rounds 3, 4, 5, the sudden-death rounds ($\geq 6$), and overall. These results, together with the transition probabilities, suggest that the round's first shooting team gets an advantage after round 2, which translates into a higher shootout win probability. This advantage seems to be particularly large when the score is tied ($s = 0$), giving Team A a higher likelihood than Team B of getting ahead in the following round.

The above calculations and statistical tests rely on the Markovian assumption, which may be restrictive. We conducted some additional data analysis to further support this assumption. The network representation presented in Fig 1 assumes that the trinomial probabilities are independent on the path at which the focal state was reached. A more general model would be to allow these probability distributions to be path-dependent, which can be specified using an ordered probit model with three outcomes ($\{-1, 0, 1\}$) that includes additional covariates capturing different paths to a state. Two such covariates where included, using an indicator variable on whether the previous rounds increased or decreased the goal difference for team A. We compared this model against the restricted Null model with no path-dependence using a likelihood ratio (LR) test. The p-value of the test is 0.53 when comparing the Null against the model with path-dependence on the preceding round (when two preceding rounds are included, the p-value of the LR test is 0.69). Based on this evidence, we cannot reject the Null that the probability transitions are indeed Markovian, which facilitates the evaluation of alternative sequences, as described in the next section.

## Alternative orders

We next estimate the influence of the fair coin for alternative orders. The ideal approach would be to run a field experiment through a real intervention, changing the order and evaluating whether it would significantly reduce the impact of the fair coin. Considering the current winning probability of 55% of Team A as the Null versus a 50% probability (no advantage) under the alternative hypothesis, approximately 600 shootouts would be required to obtain a statistical power of 80% for rejecting the Null. Such a field experiment would take about four years given the rate at which shootouts are observed in practice, which is infeasible. Hence, the second best approach is to evaluate counterfactual scenarios using historical data with the current order.

The alternative order schemes can be ranked into different levels in terms of their complexity, denoted by $x$. The current ABAB order alternates the shooter and is considered the least complex with $x = 1$. ABBA, the alternative considered by IFAB, alternates the first shooter and is one level higher in complexity, denoted $x = 2$. The Thue-Morse alternates the alterations, i.e., the alternation of AB, which is BA, is added at the end, resulting in ABBA, and then its alteration BAAB is added so we have ABBABAAB, and so on, denoted complexity $x = 3$. [12] argues that the Thue-Morse order has the potential to improve "fairness" It appears to be common to describe reducing the impact of the fair coin as being more fair. in sequential competitions such as penalty shootouts. These three order schemes are predetermined, in the sense that the order is fixed at the start of the shootout. Other designs are state-dependent, where the order of the subsequent shots also depends on the outcome of the penalties shot up to that round. See [13] for an analysis of these type of design. In this study, we focus on predetermined designs because they are simpler, but the network model can be adapted to analyze state-dependent designs too.

To facilitate the analysis, the probability network illustrated in Fig 1 gives the basis for a recursive in-play predictive model. Define $v(s, t)$ as the probability that Team A will win out of state $(s, t)$. The recursion equations for terminal states where Team A wins, terminal states where Team B wins, nodes with 2 outgoing arcs (one team may win directly out of), nodes with 3 outgoing arcs (no team can win directly out of), and nodes beyond the first 5 rounds (sudden-death rounds) are given by the following 5 expressions, respectively:

$$v(3, 4) = v(2, 5) = v(1, 6) = 1, \quad v(-3, 4) = v(-2, 5) = v(-1, 6) = 0,$$
$$v(s, t) = (1 - q_c(s, t))v(s, t + 1) + q_c(s, t)v(s + c, t + 1)$$
$$\text{for } t \in \{4, 5\}, |s| = 6 - t, c = -\operatorname{sign}(s) \cdot 1,$$
$$v(s, t) = \sum_{c \in \{-1, 0, 1\}} q_c(s, t)v(s + c, t + 1) \text{ for } s \in \{-2, \dots, 2\}, t \in \{|s| + 1, \dots, 5 - |s|\},$$
$$v(0, 6) = q_1(0, 6) + q_0(0, 6)v(0, 6).$$

In order to extrapolate predictions for the alternative order schemes, it is assumed that the transition probabilities across states in the network represented in Fig 1 depend on state $(s, t)$ alone. This assumption implies that these transition probabilities: (i) do not depend on the path leading up to round $t$ and (ii) do not depend on the within-round orders beyond round $t$. From the notation used in Fig 1, recall that when the first shooting team is ahead in the score by $s$ in period $t$, the probability of increasing this score difference in Team A's favor is denoted $q_1(s, t)$, represented by the northeast transition arc out of the node. Taking advantage of the symmetry of the network representation in Fig 1, the counterpart of this probability for the team shooting second is represented by $q_{-1}(-s, t)$, the southeast arc from node $(-s, t)$, which is the "mirror image" of node $(s, t)$ when the first team shooting in that round has the score advantage. Given our assumption, it follows that reversing the shooting order of the teams in round $t$ can be captured by using the mirror image of transition probabilities around the horizontal line in that round, i.e., by the transformation $q_c(s, t) \rightarrow q_{-c}(-s, t)$. This gives an estimate of the win probability $v_x(0, 1)$ for Team A when using order $x$. The recursion element for nodes beyond the 5th round becomes a bit different, but in the interest of brevity the details are left out. For $x = 2$ we transform the transition probabilities for $t = 2, 4, 6, 8, \dots$ (every even round), and for $x = 3$, we transform the transition probabilities for $t = 2, 3, 5, 8, \dots$ (by Thue-Morse).

Accounting for sampling error is critical to provide confidence intervals around the point estimates of the win probability under the alternative orders. Using the transition probabilities

**Table 3. Confidence bounds of the bootstrapped estimates of the win probability of Team A for alternative orders.**

| Data | Order | Win prob. | average | 1% | 5% | 50% | 95% | 99% |
|------|-------|-----------|---------|-----|-----|------|------|------|
| APH | ABAB | $v_1(0, 1)$ | 52.70% | 47.78% | 49.32% | 52.78% | 55.83% | 57.03% |
| | ABBA | $v_2(0, 1)$ | 50.60% | 46.19% | 47.60% | 50.65% | 53.40% | 54.47% |
| | T-M | $v_3(0, 1)$ | 47.41% | 42.99% | 44.37% | 47.47% | 50.21% | 51.28% |
| Ext. | ABAB | $v_1(0, 1)$ | 54.33% | 51.26% | 52.21% | 54.34% | 56.44% | 57.30% |
| | ABBA | $v_2(0, 1)$ | 50.40% | 47.71% | 48.54% | 50.40% | 52.23% | 52.98% |
| | T-M | $v_3(0, 1)$ | 47.83% | 44.70% | 45.79% | 47.87% | 49.74% | 50.50% |

of the network in Fig 1, for each bootstrapped sample ($r$) the win probability of Team A is calculated using the recursion equations for each alternative order $x \in \{1, 2, 3\}$, represented by $v_x^{(r)}(0, 1)$. The estimates $v_x^{(r)}(0, 1)$ are used to construct the distribution of the estimators and the corresponding confidence intervals.

Table 3 shows the results of the bootstrapping procedure. Columns show different statistics (average and percentiles) of the distribution of the estimators of win probabilities under the different order schemes. The estimate of the current order scheme replicates the result obtained in the analysis reported in Table 2, providing a median estimate of about 54% of winning for Team A. The 90% confidence interval with the extended dataset is [52.21%, 56.44%], therefore rejecting the Null of equal win probabilities across teams. For the $x = 2$ (ABBA) order, the point estimate is much closer to 50%, with a 90% confidence interval of [48.54%, 52.23%], and therefore we cannot reject the Null of equal win probabilities across teams. Interestingly, the $x = 3$ (Thue-Morse) order, the highest in terms of complexity, actually reverses the advantage, with a winning probability for Team A of 47.83% and a 90% confidence interval [45.79%, 49.74%]. Similar results are obtained using only APH tournaments.

It is at first surprising that Thue-Morse—which is considered the "gold standard" in fair sequential division ([14, 15])—was not the most equitable sequence in our simulations. However, it is important to consider the conditions under which Thue-Morse sequence has been reported to yield superior results: settings where the structure and parameters are stationary. For soccer penalty shootouts, there are two key deviations from this: (i) in the first two rounds there is no significant advantage of Team A, and after the second round the advantage of Team A may change between rounds; (ii) the structure itself is not stationary, as it starts with best-out-of-5 shots, and, if a winner is still not determined, switches to repetitions of best-out-of-1 shot.

## Concluding remarks

When evaluating major policy changes, the ideal analysis would be based on a randomized trial. Unfortunately, such trials are often not feasible due to the need for resources, the duration, and the cost. This paper presents the essence of a rigorous analysis that supported the decision making for a major potential policy change in the world's largest sport, namely alternative order of penalty shootouts in soccer, using historical data under the existing policy. This analysis is facilitated by a network model and an extensive dataset of penalty shootouts on the shot-by-shot level. A major benefit of the network model is that its visual representation provides an intuitive understanding and interpretation of the analysis by managers, hence building trust in the analysis. Finally, our extensive data collection helps to settle some of the controversies in previous research regarding a first-mover advantage in penalty shootouts. While being beyond the scope of a short paper, this paper invites a more in-depth study of the

mechanisms driving the impact of the shooting order. See [16] for an example of a paper with such a focus.

## Data sources

- worldfootball.net: https://www.worldfootball.net

- Linguasport: http://www.linguasport.com

- RSSSF (Rec.Sport.Soccer Statistics Foundation): http://www.rsssf.com/

- Spanish newspaper archives: http://hemeroteca.mundodeportivo.com, http://elpais.com.

- Other online sources: www.bdfutbol.com, www.wikipedia.org, www.footballdatabase.eu, www.fifa.com, www.uefa.com.

## Supporting information

**S1 File.**
(CSV)

**S2 File.**
(PDF)

**S3 File.**
(CSV)

## Acknowledgments

We thank Lukas Brud, Harry Groenevelt, Geir Jordet, Jürgen Mihm, Jo Nesbø, Petter Rudi, and Kjetil Siem for discussions and feedback. Olga Kuzmina and Elio A. Farina assisted with programming. Valentina Gorman, Angel Pavez and Kevin Ruiz manually coded data from Copa del Rey.

## Author Contributions

**Conceptualization:** Nils Rudi, Marcelo Olivares.

**Data curation:** Aditya Shetty.

**Formal analysis:** Nils Rudi, Marcelo Olivares.

**Methodology:** Nils Rudi, Marcelo Olivares, Aditya Shetty.

**Resources:** Nils Rudi, Marcelo Olivares.

**Software:** Marcelo Olivares, Aditya Shetty.

**Supervision:** Nils Rudi.

**Validation:** Nils Rudi, Marcelo Olivares, Aditya Shetty.

**Visualization:** Nils Rudi, Aditya Shetty.

**Writing – original draft:** Nils Rudi, Marcelo Olivares.

**Writing – review & editing:** Nils Rudi, Marcelo Olivares, Aditya Shetty.

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
