## [Decision Letter · Decision Letter 0]

25 May 2020

PONE-D-20-09725

Ordering sequential competitions to reduce order relevance: soccer penalty shootouts

PLOS ONE

Dear Dr. Olivares,

Thank you for submitting your manuscript to PLOS ONE.  I apologize for the delay in this review, it took some time for me to find two reviewers.

Both reviewers advise that I ask for a minor revision and that is what I would like you to do. The reviewers do not have many requests, so I would like you to respond in some way to all of them. Most important, you need to do more to meet the journal's data accessibility standards, and you need to add some discussion of the biases introduced by your modelling assumptions.

We look forward to receiving your revised manuscript.

Kind regards,

Drew Fudenberg

Academic Editor

PLOS ONE

2. Please upload a copy of Supporting Information Figure S1 which you refer to in your text on page 8.

Reviewers' comments:

Reviewer's Responses to Questions

**Comments to the Author**

1. Is the manuscript technically sound, and do the data support the conclusions?

Reviewer #1: Partly

Reviewer #2: Yes

2. Has the statistical analysis been performed appropriately and rigorously? 

Reviewer #1: Yes

Reviewer #2: Yes

3. Have the authors made all data underlying the findings in their manuscript fully available?

Reviewer #1: No

Reviewer #2: No

4. Is the manuscript presented in an intelligible fashion and written in standard English?

Reviewer #1: Yes

Reviewer #2: Yes

5. Review Comments to the Author

Reviewer #1: Summary:

This work compiles an extensive dataset of soccer penalty shootouts and conduct a counterfactual analysis in order to inform the decision by a policy unit about whether or not to alternate order of the team shooting first in each round of penalty shootouts.

The paper reviews a brief literature suggesting mixed conclusions about whether or not there is an advantage for first shooters in penalty shooters. The first shooter being drawn by a fair coin, such advantage increases the noise-to-talent ratio in what determines outcomes of soccer competitions. The paper starts by compiling a large dataset of games and confirms a statistically significant advantage of the first shooter.

Because penalty shootouts do not occur so frequently, conducting a randomized experiments powered to see if the first shooter advantage is removed by alternating the order of who shoots first across rounds would take too long. The paper takes another approach. The work estimates transition probabilities from states to states, where a state is defined by the score difference and the stage in the shootout. Under the assumption that scoring probability of each team is only influenced by these two variables at each point in time, the authors are able to simulate what would outcomes be under alternate shooting orders that are being considered by a policymaker. They find that the alternating orders would reduce the first-shooter advantage.

PLOS ONE Criteria:

(1) Primary results of original research: Yes

(2) Results not published elsewhere: OK

(3) Experiments, statistics, analyses are high-standard and described in detail: Yes, the model used is described in details and clearly. Confidence intervals are generated using a bootstrap procedure that is properly described.

Introductory discussion notes that there is no difference in the first mover advantage across competitions. Rather, each competition has a relatively small sample and thus power to detect if first shooter advantage differs across them is insufficient to make the conclusion that is does not differ. This should be rephrased but is not central for the core of the analysis.

(4) Conclusions supported by the data: Partially. The core assumptions for validity of the counterfactual are clearly stated but should be discussed further. In particular, no-path-dependence is a strong assumption that may or may not be reasonable. If Team A is leading 2-1 at the beginning of the third shootout, there is a good chance that psychological effects leading to advantages depend on whether Team B missed on the first or the second shot. The authors could discuss why they believe this assumption is reasonable and provide empirical support for this assumption using the data.

(5) Presented in intelligible fashion: Yes but this can be improved. Overall the key approach and core results are very well presented. The secondary analysis of the network, from lines 176 to lines 205 would benefit from more structure. What are the main takeaways from these various tests? Currently this part of the paper does not seem very useful because its takeaways are not explicitly connected to the key conclusions of the paper.

(6) Meets ethics standards: Yes, does not collect data from human subjects, only uses data from outcomes of public soccer games.

(7) Data availability: Authors should explain in greater details what makes them unable to provide the dataset being used. Many of the data sources being used have no restrictions (such as wikipedia) and the authors could at least provide a compiled dataset with this data, leaving out the data from sources that did not agree to share data for publication.

Conclusion

The paper makes a very clear demonstration of its approach to estimating effects of a counterfactual order in penalty shootouts. A major assumption (no path dependence in transition probabilities) is clearly acknowledged but could be a little more discussed.

The paper would also gain in interest if it could tie some of its conclusions to the cited literature on the psychology of penalty shootouts. Are there psychological reasons to explain that the Thue-Morse order reverses the Team A advantage? Are the intermediate findings from the network (currently a little disconnected from the rest of the paper), such as the fact that Team A’s advantage starts after the 2nd round, naturally related to any psychological mechanism discussed in this literature?

Reviewer #2: This paper provides evidence of the importance of order in sequential competitions in the context of penalty shootouts. The authors contribute to the current literature by compiling a comprehensive dataset of penalty shootouts, providing new estimates of the first mover advantage and constructing counterfactuals for different order sequences using a probability network model. The evidence presented is concise and compelling, the analysis of the paper is appropriate (even though the assumptions are strong) and the paper is clear and well written. The following comments may improve the presentation of the paper.

1. First, the dataset the authors compile might be of great interest to other researchers in the field, so, in the spirit of the journal, I encourage the authors to make the data publicly available. In the case that it is not possible to do so; it would be useful to provide more complete descriptive statistics. In particular, I would have found it useful to breakdown the % of Team A winning statistic by pre and post 2003 (given that the rules change), so that the numbers can be compared more directly with the literature.

2. Given that the IFAB decided not to change the system it might be worth it discussing a bit more in detail what costs are associated with changing the shootout orders. This might help inform the validity of the modelling assumptions and put the results in context. For instance, there might be a strategic reasoning to choosing which players shoot first or last that changes with the shootout order. It would also be useful to mention, briefly, what were the results of FIFA’s test for U17 and U20 tournaments.

3. In the discussion of the mixed evidence for first mover advantage in the literature review I would also mention that most players perceive shooting first as being better, as suggested by survey evidence in Apesteguia and Palacios-Huerta 2010.

4. One of the new results of the network model with round dependent states is that the first mover advantage becomes statistically significant after round 2 for tied states. It is not immediately intuitive why this would be the case. Is there a mechanism that could explain it? For example, if trainers placed their strongest players (least likely to be influenced by behavioral aspects) in rounds 1,2 and the last round, then maybe we start seeing the effect in round 3 because the players are weaker.

5. The assumption that transition probabilities depend only on the state and round number is very strong. Together with the assumption that the teams are identical this makes the counterfactual analysis highly stylized and potentially misleading as the transition probabilities computed from the data are likely to come from path dependent process and strategic considerations when changing the order might be important. As the authors point out a more in-depth analysis would be appropriate for a longer paper. However, to make the results more robust, I would add a brief discussion of what mechanisms are ruled out by the assumption (and the “mirror image” trick) and what direction this is likely to bias the results towards (my prior is that there is a downward bias when using the mirror trick). This will help contextualize why the T-M order reverts the advantage (surprising given that it is supposed to be ‘fair’), and might advocate further for not changing the order to ABBA.

6. PLOS authors have the option to publish the peer review history of their article (what does this mean?). If published, this will include your full peer review and any attached files.

Reviewer #1: No

Reviewer #2: No

---

## [Author Response · Author response to Decision Letter 0]

23 Nov 2020

We provided a detailed response to each fo the reviewers comment in the "Response to reviewers" file attached with the submission.

---

## [Editor Report · Decision Letter 1]

26 Nov 2020

Ordering sequential competitions to reduce order relevance: soccer penalty shootouts

PONE-D-20-09725R1

Dear Dr. Olivares,

Thank you for preparing a responsive revision, and for accompanying it with such a detailed explanation of how you responded to the referees' comments. I am happy to say that I am quite convinced, so I am accepting the paper without returning it to the referees .

Kind regards,

Drew Fudenberg

Academic Editor

PLOS ONE
---

## [Editor Report · Acceptance letter]

15 Dec 2020

PONE-D-20-09725R1 

Ordering sequential competitions to reduce order relevance: soccer penalty shootouts 

Dear Dr. Olivares:

I'm pleased to inform you that your manuscript has been deemed suitable for publication in PLOS ONE. Congratulations! Your manuscript is now with our production department. 

Kind regards, 

on behalf of

Dr. Drew Fudenberg 

Academic Editor

PLOS ONE